# Antiviral Potential of Plants against Noroviruses

**DOI:** 10.3390/molecules26154669

**Published:** 2021-08-02

**Authors:** Jolanta Sarowska, Dorota Wojnicz, Agnieszka Jama-Kmiecik, Magdalena Frej-Mądrzak, Irena Choroszy-Król

**Affiliations:** 1Department of Basic Sciences, Faculty of Health Sciences, Wroclaw Medical University, Chalubinskiego 4, 50-368 Wroclaw, Poland; jolanta.sarowska@umed.wroc.pl (J.S.); agnieszka.jama-kmiecik@umed.wroc.pl (A.J.-K.); magdalena.frej-madrzak@umed.wroc.pl (M.F.-M.); irena.choroszy-krol@umed.wroc.pl (I.C.-K.); 2Department of Biology and Medical Parasitology, Faculty of Medicine, Wroclaw Medical University, Mikulicza-Radeckiego 9, 50-345 Wroclaw, Poland

**Keywords:** plant secondary metabolites, antiviral activity, food, noroviruses, MNV, FCV

## Abstract

Human noroviruses, which belong to the enterovirus family, are one of the most common etiological agents of food-borne diseases. In recent years, intensive research has been carried out regarding the antiviral activity of plant metabolites that could be used for the preservation of fresh food, because they are safer for consumption when compared to synthetic chemicals. Plant preparations with proven antimicrobial activity differ in their chemical compositions, which significantly affects their biological activity. Our review aimed to present the results of research related to the characteristics, applicability, and mechanisms of the action of various plant-based preparations and metabolites against norovirus. New strategies to combat intestinal viruses are necessary, not only to ensure food safety and reduce infections in humans but also to lower the direct health costs associated with them.

## 1. Introduction

Knowledge of food viruses is not as extensive as our understanding of bacteria or fungi, the main reason for this being the difficulties in isolating, growing, and labeling the former regarding food products. Unlike many other groups of microorganisms, food-borne viruses cannot multiply in food. However, they can apparently survive food processing and storage [1]. Food contaminated with viruses can be a source of infection in consumers. Noroviruses have been associated with many recorded major food-borne viral outbreaks worldwide, while other intestinal viruses, such as the human astrovirus (HAstV), human rotavirus (HRV), sapovirus (SaV), enterovirus (EV), or Aichi virus (AiV) have been responsible for sporadic outbreaks all over the world [2].

Human noroviruses are a major cause of epidemics and periodic acute gastroenteritis worldwide. These viruses are the most common cause of food-borne diseases in the United States and Europe, entailing the societal burden of tens of billions of dollars in estimated costs of illness [3,4,5]. Globally, the incidence of food-borne norovirus infections reaches 120 million cases and 35,000 deaths per year [6]. Official reports published in 2017 and 2018 list human norovirus among the most frequently reported triggers of food-borne outbreaks. These reports show that the virus was responsible for 140 outbreaks (35% of all outbreaks) in the United States, and 211 outbreaks (7.8%) in Europe [7,8,9,10,11]. According to the RASFF report (2019), 145 outbreaks were caused by noroviruses and other caliciviruses that were found in fish and seafood, and a further 14 outbreaks relating to non-animal products were detected in the European Union [12]. According to the CDC, norovirus was the identified etiological factor of gastrointestinal complaints in 2 outbreaks out of 4 in 2020, in 8 out of 10 in 2019, and in 5 out of 11 in 2018 [13].

The transmission of the virus to humans through the consumption of contaminated food depends on various parameters, such as virus stability, food processing methods, infectious dose, and host susceptibility [14]. It is worth noting that food ingredients can protect the virus during processing and human consumption. The infectious dose of a food-borne virus is generally low, and a small number of virus particles can cause infection. Moreover, noroviruses, as food contaminants, persist in food for a long time without loss of infectivity [2]. Many control strategies that rely on the internal and external properties of the food, e.g., pH and water activity, are ineffective against these pathogens. Heat treatment is an effective way of deactivating foodborne viruses, but it can alter the organoleptic properties (e.g., color and texture) and reduce the nutrient content (e.g., protein and vitamins) of foods [15]. Currently, consumers show an increasing demand for high-quality natural food products. One of the issues is changes to the way we eat, while another is introducing raw or mildly-heat-treated foods to everyday menus: sushi, blue beef, seafood, and insects. Shellfish, fruit, and vegetables pose a serious threat to humans because they are eaten raw [16]. These foods are prone to contamination, due to the use of fecal-contaminated water for irrigation or the lack of proper personal hygiene in the people who come into contact with food [17,18].

Noroviruses belong to a group of viruses resistant to external factors. They are not sensitive to freezing, short-term heating, ionizing radiation, organic acids, preservatives and chlorine compounds, alcohols, and other detergents. At a temperature of 60 °C, their deactivation takes place only after 30 min. In their natural environment, they can remain active for several weeks or even years [19,20]. Viral infections in which the etiological factors are viruses that contaminate food can be prevented primarily by neutralizing the source of contamination during the food sanitation processes. In the context of public health, this is a significant challenge for the food industry [21,22,23]. For this reason, both deactivating the virus and maintaining high standards risk lowering the food quality characteristics, presenting a challenge for food processors. Innovative non-thermal food processing technologies, including high-pressure processing (HPP), cold plasma (CP), ultraviolet (UV) light, radiation, and pulsed electric field (PEF) treatments have been tested for food-borne virus deactivation, sensory properties, and the retained nutritional value of processed foods [14].

In recent years, intensive research has been carried out on the properties of phytochemicals with antiviral activity. Unlike chemicals, these metabolites are a safe option if used as fresh food preservatives. New strategies to combat intestinal viruses are necessary, not only to ensure food safety and reduce the number of infections in humans but also to reduce the direct health costs associated with them [5].

The aim of our study is to review the results of the latest literature describing the applicability and efficacy of various metabolites of plant origin that could be used as modern and environmentally safe agents against human food-borne noroviruses.

## 2. Characteristics of Human Norovirus

Human Norovirus (HuNoV), formerly known as the Norwalk virus, is a non-segmented, non-enveloped RNA virus belonging to the Caliciviridae family. Caliciviruses are small viruses of 30–35 nm in size, which are visible in the microscopic image as spherical particles, devoid of envelopes and spikes [24]. Noroviruses do not multiply in vitro in cell cultures. HuNoV, as well as its surrogates that are commonly used in laboratory tests, i.e., murine norovirus (MNV) or feline calicivirus (FCV), are devoid of an envelope, contain ssRNA, and show high resistance to both antimicrobial preparations and environmental conditions [14,25].

According to the latest systematics, noroviruses are divided into seven gene groups (from GI to GVII) with 30 genotypes detected globally. GI, GII and GIV are the most common causes of human infections. Many international epidemic surveillance systems (CaliciNet and NoroNet) record the transmission of norovirus infections and provide important information about the spread of different human norovirus strains. According to Hoa Tran et al. [26], the strains with the GII.4 genotype accounted for 70–80% of all the outbreaks reported over the past decade. The frequency of genotypes varied according to the population level and the route of transmission [27]. The GII.4 genotype is more commonly associated with dissemination via interpersonal contact, while non-GII.4 genotypes, such as GI.3, GI.6, GI.7, GII.3, GII.6 and GII.12, are most commonly transmitted by food [28]. Water transmissions occur more frequently among GI gene group strains than GII7 strains. This may be related to the fact that GI strains have higher water stability than GII strains [29]. Between 2009 and 2013, the GII.4 genotype was the cause of 2853 (72%) outbreaks in the USA, of which, 94% were GII.4 New Orleans or GII.4 Sydney [30].

Viruses do not multiply on the surfaces of raw food. Viral particles will not increase in number when introduced into raw food as their site of primary contamination. On the contrary, their numbers may drop over an extended period of storage, or change, subject to the conditions of their storage. Cold storage of raw produce, often at temperatures below 0°C, preserves the viruses present on them, leaving food still contaminated and therefore potentially infectious [31].

## 3. Methodology of Research Regarding the Antiviral Activity of Phytochemicals

Due to the fact that plant extracts may contain several dozen to several hundred compounds, standardization is necessary, taking into account their unique chemical profiles. In accordance with international standards, such characteristics should also include the systematic affiliation of the plant from which the oil or extract is derived and define the physicochemical properties of these phytochemicals [32]. 

The antiviral activity of essential oils and plant extracts is lower in food matrices, in comparison with in vitro tests. The lowest concentration of oils necessary to inhibit the growth of microorganisms in the food may be over 1000 times higher than are needed in the model conditions in in vitro studies [33].

To ensure that the range of activity of biologically active compounds in food is determined with precision, it is necessary to employ an adequately designed experimental analysis. A testing methodology of the antiviral activity of metabolites of plant origin must satisfy a number of criteria: for example, the starting titer must be determined correctly, according to the tested virus; the cytotoxicity of the plant product has no effect on the cell growth and/or cell morphology; the plant-derived phytochemicals in question show antiviral activity against the tested virus model [34].

Determining the antiviral effect of a biologically active preparation requires confirmation with appropriate tests. The use of the suspension method in the first stage of the research allows us to determine whether the active plant metabolite, being a component or one of the components of the tested preparation, exhibits antiviral activity [35]. In the next step, the test viruses are exposed to the plant product at different concentrations, contact times and temperatures, which allows the titer of the infectious virus to be determined. The virus’s infectious titer is determined by assessing the presence or absence of a cytopathic effect in the cell culture. The ability of the tested plant product to deactivate the test virus is determined by decreasing its infectious titer when compared to the control mixture [36].

The virucidal activity of the tested preparation against a specific virus is confirmed if the infectious virus has decreased by at least 4 logs in the titer compared to the control mixture. This means a loss of viral infectivity of 99.99% [37].

The use of cell models in in vitro tests allows for the quick and precise determination of the antiviral activity of various preparations [38]. All results of in vitro studies on the action of plant-derived active metabolites (e.g., endpoint titration technique (EPTT), virus-induced cytopathic effect inhibition (CPE), virus yield reduction assay, MTT assay, plaque reduction assay, virus deactivation assay, virus adsorption assay, virus attachment, and virus penetration assay [39]) must also be confirmed by in vivo testing [40,41,42], which, at the next stage, is a necessary step in the application for registration with government food and drug control agencies and in making the preparation available to food pharmaceutical industries [43].

The results of in vitro studies regarding the activity of plant-derived metabolites require confirmation by reference tests, also conducted in vivo, to enable an application for registration with the appropriate government food control agency before the preparation is licensed for use in the food or pharmaceutical industries.

The factors significantly affecting the antimicrobial activity of plant preparations include the activity of food enzymes, water activity, pH, temperature, and the number of microbes contaminating a given food product [44,45]. Virions present in the food matrix and in foodstuffs were found to be more resistant to the antiviral activity of plant compounds than were virions present in water [46,47].

## 4. Mechanism of the Antiviral Action of Compounds of Plant Origin 

In recent years, many laboratories around the world have engaged in research into plant extracts and their respective biological activity. Plant-derived phytochemicals exhibit various antiviral activities and employ different mechanisms of action (Figure 1) [46,48]. Individual compounds isolated from plants may show a different effect than the entire extract. Considering the fact that the effectiveness of the antimicrobial action of plant preparations is based on the mutual interaction of biologically active compounds, it is especially important to understand the structure of such molecules. Bioinformatics methods have proved to be extremely helpful in this field, making it possible to study the interactions of various low-molecular compounds with viral or cellular proteins (the so-called molecular docking). Nonetheless, wider use of plant compounds with antimicrobial activity primarily depends on determining the molecular mechanism of their action [49].

The biological and pharmacological activity of plant-derived secondary metabolites, such as polyphenols, terpenes, and alkaloids, has long been known and used in medicine. Plant antiviral phytochemicals can bind to particles on the surface of the virion, preventing target cell recognition and virus adsorption via the proper receptor (Figure 1). The blockage of receptors on the host cell surface is yet one more mechanism of action that is exhibited by phytochemicals. This consists of blocking the penetration of the virus into the cell or blocking the synthesis of viral nucleic acids. The activity of these compounds may also inhibit the synthesis and post-translational processing of viral proteins. It may also block the processes relating to the assembly of daughter virions, or the release of viral daughter particles from the host cell [50] (Figure 1).

The multiplication of viruses in the host cell is dependent on both cellular and viral factors. Plant metabolites exhibiting antiviral qualities and, therefore, finding uses in antiviral drugs specifically inhibit the multiplication of viruses without damaging the host cells [48]. Their most frequent target sites of action are the molecules found on the virion’s surface that are responsible for the recognition, adsorption, and penetration of the virus into the cell. Nucleic acids (DNA or RNA), proteins, viral RNA replicases, and reverse transcriptase have also been recognized as attractive target sites for the action of these phytochemicals [50].

Considering the potential use of essential oils and other plant extracts to combat or deactivate food-borne viruses, their antimicrobial mechanism should be analyzed first. The available literature on this topic is still scarce, especially in the group of non-enveloped viruses, which, due to their structure, constitute a difficult objective for laboratory research. Plant antimicrobial metabolites may exhibit various mechanisms of antiviral activity, which is confirmed by the results obtained by the authors of experimental studies [51,52].

In the studies conducted by Gilling et al. [53,54], the influence and mechanisms of the antiviral activity of allspice oil, lemongrass oil, citrus oil (specifically, citral), oregano oil and its main active metabolite, carvacrol, against murine norovirus (MNV) were analyzed. As part of the research, tests were carried out on the infectivity of cell cultures, protection against RNase I, binding to receptors within the host cells, and imaging in a TEM microscope was performed [53,54]. Based on the results obtained, it was found that the effectiveness of active phytochemicals varies greatly depending on the type of virus. This is confirmed by previous observations, indicating that even small differences in the structure or genome of the virus can significantly affect its susceptibility to various antiviral agents [55,56]. In turn, the results obtained by Kovač et al. [57] indicated that the essential oils obtained from hyssop and marjoram were active against enveloped HSV viruses but did not deactivate the two non-enveloped viruses that were tested (HAdV-2 and MNV-1).

In non-enveloped viruses, the capsid protects the integrity of the viral nucleic acid. Viral RNA may remain intact, while changes in the structure of the capsid may deactivate the virus [58,59]. Modification of the virus capsid is one of the mechanisms that can lead to the inhibition of the virus adsorption process, which is associated with its deactivation. In the case of MNV, the results obtained by Gilling et al. [54] suggest that, as lemongrass oil and citral bind to the viral capsid, they most likely deactivate the virus by inducing conformational changes in the capsid proteins. The magnification of the viral particles, as seen in the TEM images, indicates that oregano oil and carvacrol affect the complete loss of the integrity of the capsid [53]. Various types of structural changes within the FCV capsid, and deformations of NoV (HuNoV GII.4) and MNV-1 particles, were also found after the application of cranberry juice and grape seed extract [55,60,61]. 

The blocking of the epitopes necessary for the adsorption process in viral ssRNA allowed the observation of another instance of the mechanism of action of plant-derived compounds. Thereby, the virus lost its affinity to the receptors on the surface of the host cells and was unable to infect them. In this case, the tested plant metabolites did not damage the viral RNA [54]. The exposure of FCV-F9 and MNV-1 to pomegranate juice also reduced the infectivity of the viruses studied [60].

The phytochemicals in allspice oil have been found to be virucidal against the MNV virus. They lead to the degradation of both the capsid proteins and viral RNA [54].

Our review of the literature points to the conclusion that various plant metabolites can cause a direct virucidal effect against non-enveloped virus ssRNA by degrading the capsid or viral nucleic acid. Plant-derived compounds can also bind to the surface of the virus without destroying the proteins in the capsid and, thus, interfere with its adsorption to host cells [54,62].

## 5. Plant Preparations as Antiviral Agents against Noroviruses

The antiviral activity of plant metabolites is the subject of many scientific studies [23,41,53,54,55,56,57,58,59,60,61,62,63,64,65,66,67,68,69,70,71,72,73,74,75,76,77,78,79,80,81,82,83,84,85,86,87,88,89,90,91,92,93,94,95,96,97,98,99,100,101,102,103,104,105,106,107,108,109,110,111,112,113,114,115,116,117,118,119,120,121]. The available literature includes reports of the use of various plant extracts containing essential oils and other metabolites against viruses, including noroviruses (Figure 2). The referenced publications are classified according to the various compounds of plant origin used for testing and the active metabolites they contain (Table 1). Particular attention is paid to the antiviral efficacy of the tested plant preparations and the mechanisms of their action against noroviruses. Our review presents the most interesting and promising examples of the potential use of compounds of plant origin as antiviral phytochemicals in medicine and the food industry.

### 5.1. Effect of Essential Oils on Noroviruses

Essential oils (EOs) are volatile, aromatic substances that belong to secondary plant metabolites. The main components of essential oils are terpenes, including monoterpenes and sesquiterpenes (Table 1). Each oil may contain between a dozen and several dozen compounds of various concentrations and properties. The highly diverse chemical compositions of EOs support an extremely wide range of biological activity. The biological activity of oils and their ingredients has been the subject of many in vitro studies and a few in vivo tests. Table 1 lists the essential oils and their main ingredients used in research against norovirus surrogates, i.e., feline calicivirus (FCV) and murine norovirus (MNV).

Oregano essential oil (*Origanum vulgare*) successfully deactivated non-enveloped human norovirus surrogates—feline calicivirus (FCV) and murine norovirus (MNV) [53,67]. Gilling et al. [53] noted that the antiviral effect of 4% of oregano oil resulted in a statistically significant reduction in MNV within 15 min of exposure. The authors observed changes in virus particles under transmission electron microscopy (TEM) after 24 h of exposure to oregano oil. The treated virus particles were larger (40–75 nm) than the untreated virus particles (20–35 nm). Based on the results of a cell-binding assay, an RNase I protection assay, and TEM imaging, the authors drew conclusions regarding the mechanism of action of oregano essential oil on MNV and claimed that this oil is likely to disrupt the integrity of the virus capsid. Elizaquivel et al. [67] found significant reductions in both MNV and FCV at 4% of oregano essential oil. However, the reductions turned out to be temperature-dependent. The antiviral activity of oregano essential oil was recorded only at 37 °C, while no significant reduction was observed at 4 °C.

Gilling et al. [54] used allspice (*Pimenta dioica*) and lemongrass (*Cymbopogon citratus*) essential oils at concentrations of 2% and 4% to determine their antiviral efficacy against MNV. Lemongrass essential oil in both concentrations significantly reduced the viral infectivity of MNV within 6 h of exposure, while allspice oil was effective only at a concentration of 4% after 30 min of exposure. The authors also showed that the antiviral activity of allspice essential oil was both time- and concentration-dependent, while the effects of lemongrass essential oil were only time-dependent. The research of Gilling et al. [54] included an RNase I protection experiment to assess if the MNV capsid was degraded by lemongrass and allspice oils, and a cell-binding experiment to check if both tested oils inhibited the ability of MNV to bind to the RAW 264.7 cells. The test results obtained were suggestive of degradation of the viral capsid in the samples that were treated with the lemongrass and allspice oils. Nonetheless, the specific binding of MNV particles to host cells was unchanged after exposure to the tested essential oils, which means that they do not affect viral adsorption. The authors also used TEM to determine whether there were any structural changes to the virus particles after treatment with the oils. MNV particles exposed to allspice oil turned out to be slightly larger (from 25 to 75 nm) compared to untreated MNV (from 20 nm to 35 nm). Virus particles after treatment with lemongrass oil were much longer and had a size of 100–500 nm.

The effect of clove and *Zataria multiflora* essential oils on FCV and MNV at 4 °C and 37 °C was studied by Elizaquivel et al. [67]. The results obtained showed that the concentrations of 1% clove oil and 0.1% *Zataria* oil were effective against MNV and FCV at 37 °C.

Chung et al. [92] tested the antiviral effect of an essential oil obtained from the edible medicinal plant *Artemisia princeps* var. *orientalis*, which is popular in Korea. The active compounds in this essential oil, alpha-thujone (thujone), borneol, and camphor, were used in plaque tests against MNV-1 and FCV-F9, and 48% efficacy was observed for FCV-F9 and 64% for MNV-1 at 0.1% and 0.01% concentrations of essential oil. In addition, it was found that only α-thujone showed strong antiviral activity, while in the case of borneol and camphor, no inhibitory effect was observed against FCV-F9 and MNV-1. The authors point out the need for further research to elucidate the antiviral mechanisms of the action of the essential oil obtained from *Artemisia princeps* var. *orientalis* and alpha-thujone against FCV-F9 and MNV-1, as well as the influence of temperature on the inhibition of tested noroviruses by the active phytochemicals used in the research [92].

Kovač et al. [57] investigated the ability of essential oils derived from two aromatic plants—*Hyssopus officinalis* (hyssop) and *Thymus mastichina* (marjoram)—to deactivate non-enveloped mouse norovirus (MNV-1). No significant reduction of MNV titer was observed after treatments with hyssop and marjoram at a concentration of 0.02%.

The seeds and pericarp of *Zanthoxylum schinifolium* are widely used in Korea, China, and Japan as a spice. The antiviral activity of *Z. schinifolium* essential oil (ZSE) against the foodborne viral surrogates FCV-F9 and MNV-1 was analyzed, using the cytopathic effect test [82]. In this study, RAW 264.7 or CRFK cells were exposed to ZSE at concentrations of 0.00001%, 0.0001%, and 0.001% for 72 h. Inhibition of the cytopathic effect on CRFK or RAW 264.7 cells was not detected after the incubation of FCV-F9 and MNV-1 at all tested concentrations of ZSE. These results suggested that ZSE did not deactivate viruses.

Kim et al. [89] determined the effect of lemongrass essential oil on the infectivity and replication of MNV-1. From the plaque reduction test results, this oil was found to inhibit MNV-1, both in a time-dependent and dose-dependent manner (73.09%, using a concentration of 0.02%). It has been proven that lemongrass oil, and its main component citral, deactivate the viral coat proteins necessary for viral infection and inhibit replication of the viral genome in the host cells, which was further confirmed in in vivo studies.

### 5.2. Effect of Plant Extracts on Noroviruses

Plant extracts, which contain innumerable ingredients, are valuable sources of new and biologically active molecules with antimicrobial properties. Reports concerning the antiviral activity of plant extracts are rather limited.

Li et al. [61] tested grape seed extract (GSE) on noroviruses—murine norovirus MNV and human norovirus NoV GII.4. MNV infectivity was detected by plaque assay, while NoV GII.4 infectivity was examined by cell-binding reverse transcription-PCR, after treatment of GSE with two solutions: 0.2 mg/mL and 2 mg/mL. The infectivity of MNV was reduced to >3-log PFU/mL. The ability of NoV GII.4 to bind to the cells of the human enterocytic Caco-2 cell line was significantly reduced by treating GSE in a dose-dependent manner. The authors also checked the effect of GSE on NoV GII.4 P particles using a saliva-binding enzyme-linked immunosorbent assay. The P domain formed the outermost surface on the NoV protein capsid, and this was needed for viral binding to carbohydrate receptors on the host cells. The binding signal (OD_450_) of NoVs GII.4 P particles to the salivary carbohydrate coat on the ELISA plate was reduced. Based on the results obtained in the plaque assay for MNV-1, cell-binding RT-PCR for human NoV GII.4, and saliva-binding ELISA for human NoV GII.4 P particles, the authors concluded that GSE may cause the denaturation of viral capsid protein. Therefore, the morphology of NoV GII.4 before and after GSE treatment was examined by TEM. Human NoVs in the untreated control sample appeared as small spherical particles of two sizes: 18–20 nm and 30–38 nm. After treatment with GSE at 0.2 mg/mL, the viral particles clumped together. The deformation of most of the larger particles was also observed. At a dose of GSE of 2 mg/mL, the spherical particles disappeared, and a high concentration of residual protein was observed. These results provided direct evidence that GSE could effectively damage the NoV capsid protein.

The antiviral properties of GSE were described by Su and D’Souza [93]. They assessed GSE activity against the human norovirus surrogates, MNV-1 and FCV-F9, using lettuce and jalapeno peppers, which are frequently associated with foodborne outbreaks. Lettuce and jalapeno peppers were inoculated with MNV-1 and FCV-F9 at high (~7 log10 PFU/mL) or low (~5 log10 PFU/mL) titers, and were treated with 0.25, 0.5, 1 mg/mL GSE for 30 s to 5 min. At the higher titers, FCV-F9 was reduced by 2.33, 2.58, and 2.71 log10 PFU on lettuce, and 2.20, 2.74, and 3.05 log10 PFU on peppers after 1 min, respectively. Low FCV-F9 titers could not be detected after 1 min at all three GSE concentrations. The low MNV-1 titer was reduced by 0.2–0.3 log10 PFU on lettuce and 0.8 log10 PFU on paprika. High-titer MNV-1 was not reduced by GSE at all three tested concentrations.

The aim of the study by Joshi et al. [94] was to determine the antiviral activity of GSE against FCV-F9 and MNV-1 at both room temperature and 37 °C, and in complex food matrices within 24 h. Based on the results obtained, it was found that the antiviral effect of the tested extract increased proportionally to the time and dose. On the other hand, conducting tests in model food (apple juice and 2% milk) and simulated gastric conditions weakened the effect of GSE.

Oh et al. [95] determined the effect of mulberry seed extract (MSE) on FCV-F9 and MNV-1, using plaque assays. The antiviral effects of MSE at concentrations of 0.01, 0.1, and 1 mg/mL were assessed at various times during viral infection to assess the mechanism of antiviral action: cell pre-treatment, viral pre-treatment, concurrent treatment, and post-treatment. The maximum antiviral effect of MSE against MNV-1 and FCV-F9 was achieved when MSE at 1 mg/mL was added, along with viruses simultaneously added to RAW 264.7 and CRFK cells with viruses. The results obtained suggest that MAS may affect both noroviruses in the initial phase of viral replication. 

The ability of persimmon, wattle, coffee, and green tea extracts to deactivate FCV and MNV was tested [96]. The results showed that the persimmon extract deactivated both viruses, inhibiting their infectivity. Both wattle and green tea extracts reduced the infectivity of FCV. Coffee extract had no suppressive effect on any virus.

Other studies have shown that green tea extract (GTE) inhibits the replication of MNV and FCV [21,52,97]. Additionally, it has been observed that GTE and catechins can deactivate these viruses by non-specific binding to their receptors, thus preventing the virus from binding to host cells [97]. MNVs were completely deactivated by GTE at 37 °C [46]. Based on subsequent research results, it was also found that the accumulation of catechin derivatives during the storage of the mature green tea extract (aged-GTE) (24 h at 25 °C) resulted in a significant increase in the antiviral activity of GTE against human GII.4 norovirus under laboratory conditions [98,99]. The results obtained by Falco et al. [99] indicate a potential use of the synergistic antiviral effect of aged-GTE, and gentle heat treatment (50 °C, 30 min) to ensure food safety, mainly in fruit juices.

Randazzo et al. [100] observed a complete inhibition of human norovirus GII.4 replication by aged-GTE at concentrations of 1 mg/mL at 37 °C, 1.75 mg/mL for 21 °C, and 2.5 mg/mL at 7 °C.

Oh et al. [101] investigated the antiviral activity of methanol extracts from medicinal plants, including spices, herbal teas, and medicinal herbs, against FCV by using a plaque reduction test. Spices: garlic, ginger, red pepper; herbal teas: rosemary, green tea; and medicinal herbs: rhizome of *Cnidium*, safflower, raisin tree, trifoliate orange, danggwi and mandarin peel were used in testing. The antiviral activity of the plant extracts was measured in a plaque reduction assay in which activity was expressed as an EC50 value. Among the investigated medicinal extracts, green tea extract showed the most effective anti-FCV activity. The EC50 value was 0.13 mg/mL. Danggwi, safflower, rosemary, orange trifoliate, and tangerine peel extracts also showed antiviral activity. The EC50 values were 0.26, 0.27, 0.34, 0.49, and 0.54 mg/mL, respectively.

Other authors used aqueous extracts obtained from cloves, fenugreek, garlic, onion, ginger, and jalapeno, which were also tested for antiviral activity, using FCV as a substitute for human norovirus. Based on the test results, it was found that the use of clove extracts (eugenol—29.5%) and ginger (1,2-propanediol—10.7%) deactivated 6.0 and 2.7 logs of the initial viral titers, respectively [102].

Seo and Choi [103] determined the activity of 29 edible Korean herbal extracts against the human norovirus surrogates, MNV and FCV. Preliminary results indicate that extracts obtained from *Camellia sinensis*, *Ficus carica*, *Pleuropterus multiflorus*, *Alnus japonica*, *Inonotus obliquus*, *Crataegus pinnatifida,* and *Coriandrum sativum* showed inhibitory activity against MNV and FCV, which allows their use as natural antiviral agents. 

Park et al. [104] determined the antiviral activity of 5%, 10%, and 15% vinegar (6% acetic acid) against MNV-1 in edible, experimentally contaminated fresh seaweed (*Enteromorpha intestinalis*). After a 7-day storage period at 4 °C, a significant decrease in the MNV-1 titer was observed. In other studies, capsaicin was also found to contribute to the reduction of MNV during kimchi fermentation at various temperatures [105].

Polysaccharide-rich aqueous (HWE) and alcoholic (HEE) extract of *Houttuynia cordata*, with pharmacological properties, were used by Cheng et al. [106]. Additionally, the *H. cordata* polysaccharide (HP) with a molecular weight of ~43 kDa, which consisted mainly of galacturonic acid, galactose, glucose, and xylose, was also used to determine the antiviral potential against MNV-1. HWE was shown to be the most effective in the plaque test. HP deformed and inflated virus particles. These changes made it difficult for viruses to penetrate target cells, which confirmed the antiviral properties of HP [106].

The aim of the study by Joshi et al. [107] was to determine the antiviral activity of aqueous *Hibiscus sabdariffa* extracts against FCV-F9 and MNV-1. FCV-F9 titers were reduced to undetectable levels after 15 min at concentrations of 40 and 100 mg/mL of hibiscus extract; in the case of MNV-1, a similar effect was obtained only after 24 h.

Solis-Sanchez et al. [23] examined the antiviral effect of *Lindera obtusiloba* leaf extract (LOLE) with a significant content of pinene (49.7%), phellandrene (26.2%), and limonene (17%). These compounds significantly inhibited the infectivity of MNV-1. Preincubation of viruses with LOLE at concentrations of 4, 8, or 12 mg/mL for 1 h at 25 °C reduced the infectivity of MNV-1 by 51.8%, 64.1%, and 71.2%, respectively. The results of studies concerning the antiviral activity of LOLE, as obtained by the authors, did not make it possible to establish the mechanisms of action of these phytochemicals on the viruses studied. Further experiments are needed to clarify these issues.

### 5.3. Effect of Bioactive Plant Compounds on Noroviruses

The antiviral activity of essential oils and plant extracts may be related to the presence of bioactive compounds.

Thyme and oregano contain significant amounts of monoterpenes, such as thymol and carvacrol. Gilling et al. [53] determined the antiviral efficacy of carvacrol, which is the main active ingredient in oregano essential oil. Depending on the geographic origin, its content can be as high as 85%. Carvacrol was tested at concentrations of 0.25% and 0.5%. Both concentrations resulted in a statistically significant reduction in MNV within 15 min, in comparison with the control sample. The authors used an RNase I protection experiment and a cell-binding experiment in the study to determine the likely mechanism of carvacrol’s action on MNV. The reductions observed in cell culture infectivity for carvacrol increased with greater durations of exposure to carvacrol (e.g., from 1.28-log10 after 15 min to >4.52-log10 after 24 h of exposure to the 0.5% concentration), whereas the reductions observed in the viral RNA were initially greater. These results suggest that carvacrol partially degraded the capsid, but the virus may still be infectious. TEM images showed that all the carvacrol-treated virus particles were greatly expanded in size (100 to 900 nm). Among them were both intact particles and others completely broken into capsid components.

Carvacrol at various concentrations (0.25, 0.5, 1% for 2 h at 37 °C) was used in the MNV and FCV deactivation test at titers of about 6–7 log TCID50/mL. Carvacrol, at a concentration of 0.5%, completely deactivated both norovirus surrogates. In addition, it was also found that 0.5 or 1% carvacrol can be used in lettuce-washing water to reduce the MNV and FCV titer, which indicates the possibility of using this plant metabolite as a natural viral contamination-reducing agent in fresh vegetables [108].

Thymol was also effective in reducing the titer of norovirus surrogates in a dose-dependent manner. Thymol in concentrations of 0.5 and 1% reduced FCV titers to undetectable levels, while in the case of MNV, thymol at concentrations of 1 and 2% reduced them by 1.66 and 2.45 log TCID50/mL, respectively [109].

Antiviral activity against MNV-1 was also found using natural extracts of *Aloe vera* and *Eriobotryae folium*. Aloin and emodin are the main active metabolites of both extracts [110].

The antiviral activity of citral, one of the main active ingredients of lemongrass oil, was studied by Gilling et al. [54]. Both 2% and 4% citral concentrations significantly reduced the infectivity of the MNV cell cultures over 6 and 24 h of exposure, compared to controls. The citral-treated MNV particles were greatly enlarged to an average size of 600 nm. However, the citral-treated MNV particles appeared intact.

Catechins are an important active ingredient in green tea. The antiviral activity of four catechins—epigallocatechin (EGC), epicatechin (EC), epigallocatechin gallate (EGCG), and epicatechin gallate (EKG)—was determined by Oh et al. [101]. EGCG, which is the main component of green tea, showed the most effective activity (EC_50_, 12 mg/mL) against FCV.

The effect of cranberry proanthocyanidins (PAC) at concentrations of 0.30, 0.60 and 1.20 mg/mL on MNV and FCV was determined [55]. At low viral titers (~5 log10 PFU/mL), FCV was undetectable after 1 h of exposure to the three tested PAC solutions, while the MNV decreased by 2.63, 2.75 and 2.95 log10 PFU/mL from 0.15, 0.30 and 0.60 mg/mL PAC, respectively. Experiments with high viral titers (~7 log10 PFU/mL) showed similar trends but with reduced effects. Su et al. [111] showed that the viral reduction within the first 10 min of PAC treatment was ≥50% of the total reduction. Structural changes in PAC-treated FCVs were observed under TEM.

Su et al. [112] investigated the effect of pomegranate polyphenols on the infectivity of FCV and MNV. Viruses with high (~7 log10 PFU/mL) or low (~5 log10 PFU/mL) titers were treated with pomegranate polyphenols at concentrations of 8, 16, and 32 mg/mL. FCV was undetectable after 1 h of exposure to all pomegranate polyphenols tested, using both low and high titer. MNV with low initial titers decreased by 1.30, 2.11, and 3.61 log10 PFU/mL, and at high initial titers by 1.56, 1.48, and 1.54 log10 PFU/mL, respectively, from treatment with 4.8 and 16 mg/mL of pomegranate polyphenols. Su et al. [60] described the time-dependent effect of pomegranate polyphenols at two concentrations (2 and 4 mg/mL) on the infectivity of FCV and MNV. The reduction of viral titer by pomegranate polyphenols was found to be a rapid process, with a ≥50% reduction in titer within the first 20 min of treatment. The FCV and MNV-1 titers were reduced by 4.02 and 0.68 log10 PFU/mL at 2 mg/mL pomegranate polyphenols. In the presence of pomegranate polyphenols at a concentration of 4 mg/mL, the FCV and MNV titers decreased by 5.09 and 1.14 log10 PFU/mL, respectively.

The antiviral activity of myricetin, L-epicatechin, tangeretin and naringenin, belonging to the flavonoids, was established by Su et al. [113]. Flavonoids at concentrations of 0.25 and 0.5 mM were used in the research. Myricetin was found to be most effective against FCV. Low-titer FCV (~5 log10 PFU/mL) decreased to undetectable levels after treatment for 2 h with myricetin, at both 0.25- and 0.5-mM concentrations. The high titer of FCV (~7 log10 PFU/mL) was reduced by 1.73 and 3.17 log10 PFU/mL with 0.25 and 0.5 mM myricetin, respectively. L-epicatechin was less effective; at 0.25 and 0.5 mM, it reduced a high-FCV titer by 0.18 and 0.72 log10 PFU/mL and a low-FCV titer by 0.33 and 1.40 log10 PFU/mL, respectively. Tangeretin and naringenin, at both concentrations tested, did not cause any significant deactivation of both high- and low-FCV titers. All flavonoids tested at 0.25 mM showed no measurable deactivation of the low-MNV titer after 2 h of incubation. Only myricetin and 0.5 mM L-epicatechin showed a negligible reduction in low-titer MNV of 0.22 log10 PFU/mL and 0.27 log10 PFU/mL, respectively. Tangeretin and naringenin at 0.5 mM showed no measurable effect on the low-MNV titer. Su et al. [113] described the effect of myricetin, L-epicatechin, tangeretin, and naringenin at concentrations of 0.25 mM on the virus adsorption and replication of FCV and MNV. Only myricetin showed a slight measurable effect on FCV adsorption to host cells. No measurable effect of all tested flavonoids on the adsorption of MNV into host cells was observed. None of the flavonoids had any effect on virus replication.

The effect of tannin-containing compounds (glucose pentagalloyl (PGG), propyl gallate (PRG), pyrogallol (PYG)) on FCV and MNV was investigated [96]. The antiviral test was performed by measuring the infectivity of the virus after treatment with tannins by the standard TCID50 method. The results obtained indicated that PGG, PRG, and PYG had a weak damping effect on FCV and MNV.

Turmeric, as an active plant component, contains 1–5% phenolic components. The antiviral properties of curcumin have been demonstrated in the example of noroviruses. Out of 18 phytochemicals used in the study, curcumin showed the most effective neutralizing activity against MNV. The action of curcumin depends on both its concentration and the time of its incubation with pathogens. The increase in the concentration of curcumin and the extension of the incubation time resulted in an increase in the amount of neutralized MNV. The studies used curcumin concentrations at 0.25, 0.5, 0.75, 1 and 2 mg/mL. The presence of curcumin at a concentration of 2 mg/mL neutralized approximately 91% of the MNV particles. Moreover, it was found that curcumin did not inhibit viral RNA replication [34].

Another study investigating the effects of curcumin on noroviruses was based on photodynamic therapy. This method consists of the production of reactive oxygen species with the participation of light-induced photosensitizers [114]. One of these was found to be curcumin, the effect of which on FCV and MNV was assessed after initial photoactivation with an LED diode. Although antiviral activity was found against both tested viruses, it was slightly lower for MNV. These results indicate the possibility of using photoactivated curcumin as a natural additive in the food industry, to reduce food contamination with intestinal viruses [115].

Complete inhibition of virus multiplication was observed using the extract and its fraction at concentrations of 0.1–1 mg/mL [116]. Enlarged viral capsids were observed using TEM, which could interfere with the binding of the viral surface protein to host cells. Additionally, two RCS-F1-derived polyphenolic compounds were identified that inhibited replication of the tested viruses. Test results obtained by Lee et al. [116] indicate the possibility of using black raspberry seed extract in food preservation processes.

Joshi et al. [117] assessed the antiviral effect of blueberry proanthocyanidins (B-PAC) in food matrices (apple juice and 2% milk), under simulated gastrointestinal conditions, against FCV-F9 and MNV-1. Milk, which was a much more complex food matrix compared to apple juice, inhibited the antiviral activity of B-PAC.

The results obtained by Kim et al. [41] demonstrate the inhibitory effect of fucoidans obtained from three species of brown algae (*Laminaria japonica*, LJ), *Undaria pinnatifida* (UP), and *Undaria pinnatifida* sporophyll (UPS) against MNoV, FCV, and HuNoV. The use of these compounds at a concentration of 1 mg/mL showed high antiviral activity, with a mean log decrease in viral titer of 1.1 in the plaque assays. LJ showed the greatest antiviral effectiveness (54–72% inhibition at 1 mg/mL). It was observed that pre-treatment with fuconaids interfered with the attachment of the virus to the host cell receptors. It is worth noting that, according to the authors, this is the first report in which, in in vivo studies performed on mice administered with brown algae fucoidans, a 0.6 log reduction in the MNoV titer was observed, with a corresponding improvement in the survival rates of the mice in the study group compared to the animals from the control group [41].

### 5.4. The Effect of Juices on Noroviruses

The aim of the research conducted by Horm and D’Souza [118] was to determine the survival of human MNV-1 and FCV-F9 norovirus surrogates in orange and pomegranate juices, and a mixture of both juices, over 0.1.2, 7, 14, and 21 days in a refrigerator (4 °C). Both juices were inoculated with each virus for 21 days, then serially diluted in a cell culture medium, and plaques were tested. MNV-1 showed no titer reduction after 21 days in orange juice. A moderate reduction in titer (1.4 log) was found in the pomegranate juice. MNV-1 was completely reduced after 7 days in a mixture of orange and pomegranate juice. FCV-F9 was completely reduced after 14 days in orange and pomegranate juice. FCV-F9 was completely reduced after 1 day in a mixture of orange and pomegranate juice.

Su et al. [112] investigated the effect of pomegranate juice (PJ) on MNV-1 and FCV-F9. Viruses with high (~7 log10 PFU/mL) or low (~5 log10 PFU/mL) titers were mixed with equal volumes of PJ and incubated for 1 h at room temperature. Post-treatment viral infectivity was assessed using standard plaque tests. PJ lowered the FCV-F9 and MNV-1 titers by 2.56 and 1.32 log10 PFU/mL, respectively, for low titer, and 1.20 and 0.06 log10 PFU/mL for high titer, respectively. The same research group [60] determined the time-dependent effect of PJ on the infectivity of food-borne replacement viruses. Each virus at ~5 log10 PFU/mL was mixed with equal volumes of PJ and incubated for 0, 10, 20, 30, 45, and 60 min at room temperature. Reduction of the viral load by PJ was found to be a rapid process. Test viruses were reduced by ≥50% during the first 20 min of treatment. The titer decreased by 3.12 and 0.79 log10 PFU/mL, respectively, for FCV-F9 and MNV-1.

The effect of cranberry juice (CJ) on MNV-1 and FCV-F9 was studied by Su et al. [55]. Both viruses with high (~7 log10 PFU/mL) and low (~5 log10 PFU/mL) titers were mixed with equal volumes of CJ (pH 2.6) and incubated for 1 h at room temperature. The standardized plaque assay was used to assess viral infectivity. CJ reduced FCV-F9 at low viral load to undetectable levels in the suspension test, and MNV-1 decreased by 2.06 log10 PFU/mL. Experiments with high viral titers showed similar effects. In another time-dependent study by Su et al. [111], FCV-F9 at low viral titers was reduced by ~5 log10 PFU/mL, over 30 min when treated with CJ (pH 2.6 and pH 7.0). MNV-1 titers similarly decreased for CJ at pH 2.6 or 7.0.

*Rubus coreanus* is a species of black raspberry, one that is rich in polyphenols and with anti-inflammatory, antibacterial, and antiviral properties. Oh et al. [119] compared the antiviral activity of *R. coreanus* juice (black raspberry juice, BRB) and cranberry, grape, and orange juices using plaque tests. Out of all the juices tested, BRB juice was the most effective in reducing plaque formation in MNV-1 and FCV-F9. The studies attempted to determine the mechanism of action of BRB juice on viruses. The maximum antiviral effect of BRB juice on MNV-1 was observed when it was added to the cells of murine macrophage leukemic monocytes (RAW 264.7) simultaneously with the virus (co-treatment). Pre-treatment of Crandell Reese Feline Kidney (CRFK) cells or FCV-F9 with BRB juice showed significant antiviral activity. On the basis of the obtained results, it can be concluded that inhibition of viral infection with BRB juice on MNV-1 and FCV-F9 probably occurs during the internalization of virions into the cell or upon the attachment of the viral surface protein to the cell receptor.

The aim of the research carried out by Joshi et al. [120] was to determine the antiviral effect of blueberry juice (BJ) and proanthocyanidins (BB-PAC) against FCV-F9 and MNV-1 (37 °C, 24 h) by reducing plaque tests. The prophylactic and therapeutic potential of commercially available juices and BB-PAC were tested in a dose- and time-dependent manner. Based on the results obtained, it was found that both BB-PAC and BJ had an influence on the processes of adsorption and replication of the intestinal viruses studied in vitro (a reduction of MNV-1 titer to undetectable levels was observed after 3 h for 1, 2, and 5 mg/mL BBPAC, and after 6 h for BJ). Determining their antiviral activity in the presence of food matrices under simulated gastric conditions is a prerequisite for the use of these preparations in therapy [120].

The antiviral activity of *Morus alba* (mulberry juice, MA) on MNV-1 and FCV-F9 was tested by cytopathic inhibition, platelet reduction, and RNA expression assays [121]. MA juice was found to be effective in reducing the infectivity of both viruses during both initial and concomitant treatment. Juice concentrations of 0.005% (equivalent to 100% natural juice) for MNV-1 and 0.25% for FCV-F9 caused a 50% decrease in viral load. 0.1% MA juice showed approximately 60% reduction in MNV-1 polymerase gene expression, confirming suppression of viral replication. It can therefore be concluded that MA juice can inhibit MNV-1 replication and the internalization of both tested viruses.

## 6. Practical Application of Metabolites of Plant Origin in the Food Industry

One of the most effective strategies being developed in modern methods of food preservation is the application of active packaging containing essential oils. Biologically active phytochemicals are an integral component of the packaging material [122]. The active packaging interacts with the food, limits the growth of microorganisms, and deactivates viruses. In this way, active packaging largely eliminates the risk to public health and extends the shelf life of food products [123].

In recent years, intensive work has been carried out on the use of edible films and coatings with the addition of essential oils for food preservation. The advantage of this method has been demonstrated in experimental studies of contaminated fruit, vegetable, cheese, meat, and fish, where both naturally occurring microbial contaminants and artificially introduced strains were included.

Fabra et al. [124] developed antiviral active edible membranes by adding lipids to alginate membranes. The polymer matrix prepared in this way was enriched with two natural extracts with a high phenolic compound content, green tea extract (GTE) and grape seed extract (GSE). All of these are biologically active plant metabolites and, as such, showed antiviral activity against mouse norovirus (MNV). Edible antiviral coatings benefiting from the synergistic effect of carrageenan and GTE are also an innovative strategy used to eliminate or reduce the viral contamination of berries without significantly changing their physicochemical properties [125]. In addition, it was observed that GTE solutions significantly increased their antiviral activity against MNV if left in different pH conditions for 24 h. This may be related to the formation of catechin derivatives during the storage of this preparation [52]. Additionally, it was observed that GTE solutions significantly increased their antiviral activity against MNV if left in different pH conditions for 24 h, which was associated with the formation of catechin derivatives during the storage of this preparation [98].

It was also found that the addition of aging GTE to mildly heat-treated juices increased the deactivation of MNV-1 by over 4 logs. The synergistic action of both antiviral agents reduced the infectivity of MNV-1, which confirms the hypothesis that GTE can be used as an additional control agent that improves food safety [99].

The antiviral effect of *Lindera obtusiloba* leaf extract (LOLE) on MNV-1, stemming from the synergistic action of several compounds with pinene as the key molecule, was tested on fresh lettuce, cabbage, and oysters. An hour-long incubation at 25 °C with LOLE at a concentration of 12 mg/mL resulted in a significant reduction of the viral plaques (plaque formation) of MNV-1 in lettuce (76.4%), cabbage (60.0%), and oysters (38.2%). The results of these studies suggest that LOLE can deactivate norovirus and can be used as a natural disinfectant and preservative in fresh food products [23].

Antiviral activity was also found by analyzing the effects of natural *Aloe vera* and *Eriobotryae folium* extracts. Aloin and emodin, the main active phytochemicals in the extracts of these plants, showed a preservative effect. This was confirmed, based on the results of studies in which fresh cabbage was inoculated with MNV-1 on its surface [110].

Chitosan films supplemented with green tea extract (GTE) can also be applied as active packaging materials. Chitosan is a non-toxic polysaccharide polymer that is used as an ingredient in edible packaging films, where its antimicrobial activity is used to increase the shelf life of food products. Natural plant metabolites with antimicrobial activity, e.g., essential oils and plant extracts, may be considered as possible components of edible films. It is important that all the above phytochemicals have GRAS (Generally Recognized as Safe) status. It was found that, after 24 h of incubation with the addition of 5 and 10% GTE, there was a significant reduction in the MNV-1 titer by 1.6 and 4.5 logs, respectively. Films containing 15% GTE reduced MNV-1 to undetectable levels [21].

The encapsulation of essential oils (capsules with a size of 1–1000 μm (microcapsules) or 1–100 nm (nanocapsules) offers another opportunity to preserve food using essential oils [122]. Polyethylene, carbohydrates (starch, cellulose, chitosan), proteins (casein, albumin, gelatin), fats (fatty acids, waxes, paraffin), and gums (alginates, carrageenan, acacia) are the materials most often used in this technology. Essential oils enclosed in capsules maintain greater stability, and this determines their optimal antimicrobial properties [126].

## 7. Summary

Noroviruses are highly resistant to environmental factors, so they can be efficiently transmitted through food, water, or surfaces contaminated with them, and pose a potential threat to public health. Antiviral metabolites of plant origin have important advantages over synthetic preservatives used as fresh food disinfectants because they are effective at safe dosages, are generally available, and use the inability of microorganisms to become resistant to plant-based viroids. As secondary metabolites of plants, essential oils, and plant extracts are part of their defense system against pathogens. Therefore, they often exhibit antimicrobial, including antiviral, activities. The activity spectrum of plant metabolites is diverse. The effectiveness of plant preparations and the possibility of their use in fighting intestinal viruses such as noroviruses is primarily dependents on the qualitative and quantitative composition of biologically active phytochemicals, and their concentration in food.

## Figures and Tables

**Figure 1 molecules-26-04669-f001:**
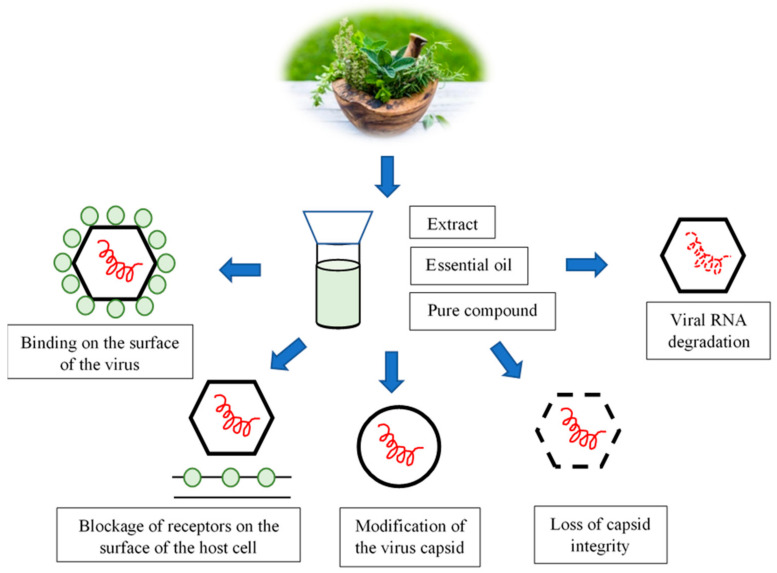
Representation of different possible modes of action of plant extracts, essential oils, and their constituents against noroviruses.

**Figure 2 molecules-26-04669-f002:**
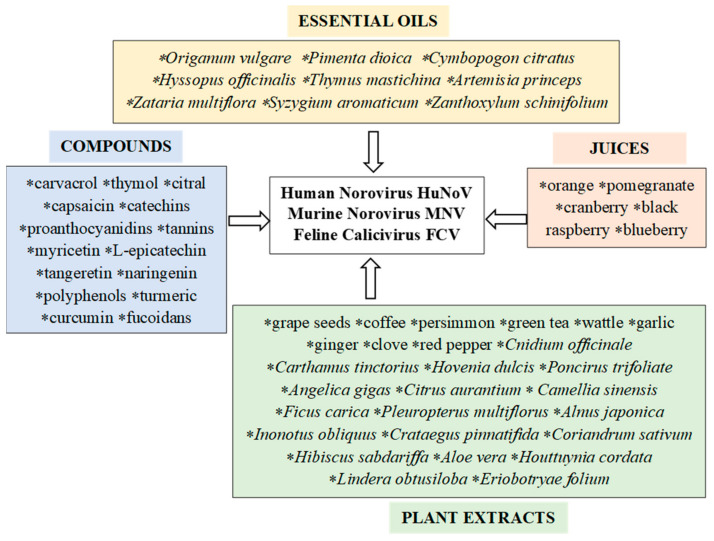
Antiviral activity of metabolites of plant origin against noroviruses.

**Table 1 molecules-26-04669-t001:** The composition of essential oils and their antiviral activity against noroviruses.

Essential Oil	Plant	Main Constituents	Group of Chemical Compounds	Content (%)	Viruses	References
Oregano	*Origanum vulgare*	Carvacrol	Monoterpene	0.3–80.8	MNV FCV	[53,63,64,65,66,67]
		Thymol	Sesquiterpene lactone	0.96–63.7		
		P-cymene	Related to monoterpene	<0.1–16.94		
		Gamma-terpinene	Monoterpene	0.8–21.0		
		Alpha-terpineol	Monoterpene alcohol	<0.09–12.0		
		Limonene	Monoterpene	0.3–0.7		
Marjoram	*Thymus mastichina*	Linalool	Monoterpene alcohol	24.5–73.5	MNV-1	[57,63,68,69,70]
		1,8-cineole	Monoterpene deriv.	9.4–55.6		
		Beta-pinene	Monoterpene	0.6–5.9		
		Alpha-pinene	Monoterpene	0.9–4.3		
		Alpha-terpineol	Monoterpene alcohol	0.9–3.0		
		Camphor	Monoterpene deriv.	0.00001–3.0		
		Limonene	Monoterpene	0.4–2.1		
Thyme	*Thymus vulgaris*	Thymol	Sesquiterpene lactone	27.6–100	MNV-1	[57,63,71,72]
		Trans-sabinene hydrate	Monoterpene hydrate	0.43–39.4		
		Menthol	Monoterpene alcohol	1.3–39		
		Bornyl acetate	Monoterpene	0.2–25.57		
		Limonene	Monoterpene	0.4–24.2		
		Carvacrol	Monoterpene	2.0–20.5		
		Gamma-terpinene	Monoterpene	0.6–14.9		
*Zataria multiflora*	*Zataria multiflora* Boiss.	Thymol	Sesquiterpene lactone	40.8	MNV FCV	[63,67,73,74,75,76,77]
		Carvacrol	Monoterpene	27.8		
		Ρ-cymene	Related to monoterpene	8.4		
		Gamma-terpinene	Monoterpene	4.0		
		Beta-caryophyllene	Sesquiterpene	2.0		
		Linalol	Monoterpene alcohol	1.7		
		Alpha-terpinolene	Monoterpene	1.3		
Clove	*Syzygium aromaticum* (*Eugenia caryophyllus*)	Eugenol	Monoterpene deriv.	86.7	MNV FCV	[63,67,78,79]
		Beta-caryophyllene	Sesquiterpene	3.2		
		Allo-aromadendrene	Sesquiterpene	1.3		
		Alpha-humulene	Sesquiterpene	0.9		
Hyssop	*Hyssopus officinalis*	Linalool	Monoterpene alcohol	49.6	MNV-1	[57,63,80,81]
		1,8-cineole	Monoterpene deriv.	13.3		
		Limonene	Monoterpene	5.4–12.2		
		Beta-pinene	Monoterpene	3.0–11.1		
		Beta-caryophyllene	Sesquiterpene	1.5–2.8		
		Isopinocamphone	Bicyclic monoterpenoids	1.3–43.3		
*Zanthoxylum schinifolium*	*Zanthoxylum schinifolium*	Estragole	Phenylpropene	42.0	MNV-1 FCV-F9	[63,82,83]
		Oleic acid	Monounsaturated omega-9 fatty acid	20.97		
		Palmitic acid	Saturated fatty acid	19.86		
		2,4-Decadienal	Polyunsaturated fatty aldehyde	4.87		
		2-Undecenal	Aldehyde	3.81		
Allspice	*Pimenta dioica*	Eugenol	Monoterpene deriv.	45.4–83.68	MNV	[54,63,84,85,86]
		Beta-caryophyllene	Sesquiterpene	2.3–8.9		
		P-cymene	Related to monoterpene	1.77–1.78		
		Terpinolene	Monocyclic monoterpene	1.23–2.35		
		Alpha-cadinol	Pseudoguaianolide	1.0–5.9		
		Alpha-humulene	Sesquiterpene	0.88–5.4		
Lemongrass	*Cymbopogon citratus*	Geranial	Monoterpene aldehyde	32.7–49.9	MNV-1	[54,63,87,88,89]
		Neral	Monoterpene aldehyde	26.5–38.2		
		Myrcene	Monoterpene	1.7–25.3		
		Nerol	Monoterpene	0.2–12.5		
		Geraniol	Monoterpene deriv.	0.2–10.4		
		1,8-cineole	Monoterpene deriv.	0.2–2.9		
Tea tree	*Artemisia princeps* var. *orientalis*	1,8-cineole	Monoterpene deriv.	2.2–24.3	MNV-1 FCV-F9	[63,90,91,92]
		Borneol	Monoterpene alcohol	2.1–5.6		
		Camphor	Monoterpene deriv.	1.4–38.7		
		α-terpineol	Monoterpene alcohol	1.1–9.8		
		Beta-pinene	Monoterpene	0.6–11.7		
		Alpha-pinene	Monoterpene	0.5–9.7		
		Beta-caryophyllene	Sesquiterpene	0.4–10.6		
		Isoborneol	Monoterpene deriv.	0.1–20.9		
		Alpha-thujone	Monoterpene	0.1–16.0		

## Data Availability

Not applicable.

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
