# Peer review of "Antiviral Potential of Plants against Noroviruses"

_molecules, 2021, doi:10.3390/molecules26154669_

Round 1

Reviewer 1 Report

I reviewed manuscript „Methods of obtaining and potential possibilities of using plant secondary metabolites in food as agents against noroviruses – the latest reports“

I am not sure what mean methods of obtaining. It will be much better that authors just focused in the manuscript on Possibilities of using essential oils as agents against noroviruses. But in most of the manuscript they actually focused on viruses and microbes at whole, not particularly on noroviruses.

It is obvious that different manuscript parts were written by different authors and all together they do not represent the meaningful whole. Although the topic is interesting the manuscript is very poorly prepared.

There are no authors addresses, abstract and keywords in the main manuscript.

The lines are not numbered and it is difficult to review such a poorly technically prepared manuscript.

I am not sure that all statements in an Introduction are properly cited. For example, first and second paragraphs in manuscript contain a lots of statements but just one references at the end of each paragraph. Similar is across the manuscript. In some part looks like that sentences are generated by software who mixed together abstracts from different papers.

Introduction section are in fact not introduction, it is section about noroviruses. Introduction should contain information about the whole manuscript, background and why authors decide to prepare this manuscript.

Section 2. Methods of obtaining biologically active compounds from plant materials very hard to follow. Authors mention several ways of extraction, and the whole section is mixed up. They start with essential oils and they all mixed up, they did not explained methods and did not mention all ways of extraction. Also every paragraph contains one citation and they are not related. I have a long year experience with extraction of bioactive compounds from plants but I am not able to understand what authors say in this paragraph.

Section 4. Methodology of research on the antiviral activity of substances of plant origin should be before section 3. Mechanism of antiviral action of compounds of plant origin. Probably scientist first used methodology and then discovered mechanisms.

„Essential oils and plant extract in vitro tend to be active at much lower concentrations than in food matrices “sentences not clear.

At the end in section 4. authors did not explain the methods which can be used in an antiviral testing. The principles, test development etc. Section contain several unrelated paragraphs.

First paragraph in section 5. „Essential oils (EO) are products of the secondary metabolism of plants. The most important components of essential oils are organic substances - terpenes. These com-pounds are natural oligomers of isoprene widespread in nature. Essential oils are con-cent rated natural products with a strong scent. They are a mixture of many chemical compounds, mainly terpenes, especially monoterpenes and sesquiterpenes, diterpenes, although they can also be present together with other molecules such as esters, alcohols, aldehydes, ketones, phenols, ethers, and hydrocarbons (Table 1).“ is wrong.

Firstly, EO are mixtures of secondary metabolites mostly terpenoids, but monoterpenoids and diterpenoids, not oligomers of isoprene.

I am actually not sure how section 5. Composition and biological activity of essential oils is related with the topic of review.

Section 6. Review of the literature on research into substances of plant origin with potential antiviral activity targeting human noroviruses is actually what the whole manuscript should contain. I am not sure what represents section 1-5.

Section 6 is also full of repetition. Authors more than 10 times in the manuscript mention what are essential oils and every times they use different definition, sometimes completely wrong one. Section is written using unrelated paragraphs, and looks like that author just copy-paste abstracts from different authors, without commenting results or connected them in any way.

Author Response

Dear Reviewer,

We would like to thank you very much for your insightful reviews of our manuscript. Thank you for all your comments and suggestions. Thank you for your taking the time.

I reviewed manuscript „Methods of obtaining and potential possibilities of using plant secondary metabolites in food as agents against noroviruses – the latest reports“

  1. I am not sure what mean methods of obtaining. It will be much better that authors just focused in the manuscript on Possibilities of using essential oils as agents against noroviruses. But in most of the manuscript they actually focused on viruses and microbes at whole, not particularly on noroviruses.

Author response:

We agree with the reviewer's suggestion. We believe that Section 2 (Methods of obtaining biologically active compounds from plant materials) is redundant because it departs from the subject of our review. We have decided to delete this chapter from the manuscript.

As suggested by the reviewer, we have also removed the information on the antibacterial and antifungal effects of plant substances, focusing solely on noroviruses (Table 1).

  1. It is obvious that different manuscript parts were written by different authors and all together they do not represent the meaningful whole. Although the topic is interesting the manuscript is very poorly prepared.

Author response:

The manuscript was reread by all authors and edited.

  1. There are no authors addresses, abstract and keywords in the main manuscript.

Author response:

We apologize for this oversight. Our manuscript was not prepared in the Microsoft Word template format. Now, the revised manuscript has been prepared in this format and contains all the required data.

  1. The lines are not numbered and it is difficult to review such a poorly technically prepared manuscript.

Author response:

We apologize for this oversight. In the current version of the manuscript, the lines are numbered (the Microsoft Word template format has been used).

  1. I am not sure that all statements in an Introduction are properly cited. For example, first and second paragraphs in manuscript contain a lots of statements but just one references at the end of each paragraph. Similar is across the manuscript. In some part looks like that sentences are generated by software who mixed together abstracts from different papers.

Author response:

The citations have been completed in the revised version of the manuscript.

The authors did not use any software for mixing abstracts or other similar applications. The authors prepared the manuscript fragments alone focusing on the most important results of the experiments obtained by different research groups.

  1. Introduction section are in fact not introduction, it is section about noroviruses. Introduction should contain information about the whole manuscript, background and why authors decide to prepare this manuscript.

Author response:

We agree with the reviewer's suggestion.

The introduction section has been changed. However, the necessary information about noroviruses was provided in a separate section entitled “Characteristics of Human Norovirus”

  1. Section 2. Methods of obtaining biologically active compounds from plant materials very hard to follow. Authors mention several ways of extraction, and the whole section is mixed up. They start with essential oils and they all mixed up, they did not explained methods and did not mention all ways of extraction. Also every paragraph contains one citation and they are not related. I have a long year experience with extraction of bioactive compounds from plants but I am not able to understand what authors say in this paragraph.

Author response:

We agree with this suggestion.

Section 2 titled Methods of obtaining biologically active compounds from plant materials) is redundant because it differs from the topic discussed in our article.

  1. Section 4. Methodology of research on the antiviral activity of substances of plant origin should be before section 3. Mechanism of antiviral action of compounds of plant origin. Probably scientist first used methodology and then discovered mechanisms.

Author response:

We agree with this suggestion. It has been done.

  1. „Essential oils and plant extract in vitro tend to be active at much lower concentrations than in food matrices “sentences not clear.

Author response:

This sentence has been changed: “The antiviral activity of essential oils and plant extracts is lower in food matrices in comparison with in vitro tests.”

  1. At the end in section 4. authors did not explain the methods which can be used in an antiviral testing. The principles, test development etc. Section contains several unrelated paragraphs.

Author response:

Section 4 titled Methodology of research on the antiviral activity of substances of plant origin has been revised (now this is section 3). A paragraph has been added to the end of this section listing the names of the tests that are used in antiviral research.

  1. First paragraph in section 5. „Essential oils (EO) are products of the secondary metabolism of plants. The most important components of essential oils are organic substances - terpenes. These com-pounds are natural oligomers of isoprene widespread in nature. Essential oils are con-cent rated natural products with a strong scent. They are a mixture of many chemical compounds, mainly terpenes, especially monoterpenes and sesquiterpenes, diterpenes, although they can also be present together with other molecules such as esters, alcohols, aldehydes, ketones, phenols, ethers, and hydrocarbons (Table 1).“ is wrong.

Author response:

Section 5 describing the antibacterial and antifungal effects of essential oils has been removed as inconsistent with the main topic of the publication.

Table 1 has been modified. Currently, it contains information on noroviruses.

  1. Firstly, EO are mixtures of secondary metabolites mostly terpenoids, but monoterpenoids and diterpenoids, not oligomers of isoprene.

Author response:

This is true, we agree with the reviewer. This was changed in the manuscript text.

  1. I am actually not sure how section 5. Composition and biological activity of essential oils is related with the topic of review.

Author response:

Section 5 has been deleted.

  1. Section 6. Review of the literature on research into substances of plant origin with potential antiviral activity targeting human noroviruses is actually what the whole manuscript should contain. I am not sure what represents section 1-5.

Author response:

The manuscript has been checked and corrected. Its current version contains sections related only to noroviruses.

  1. Section 6 is also full of repetition. Authors more than 10 times in the manuscript mention what are essential oils and every times they use different definition, sometimes completely wrong one. Section is written using unrelated paragraphs, and looks like that author just copy-paste abstracts from different authors, without commenting results or connected them in any way.

Author response:

Section 6 (now Section 5) has been corrected and completed.

The content of this section has been divided into subsections: 5.1. Effect of essential oils on noroviruses, 5.2. Effect of plant extracts on noroviruses, 5.3. Effect of bioactive plant compounds on noroviruses, 5.4. The effect of juice on noroviruses. In this way, we wanted to systematize the knowledge about the effects of individual products of plant origin on noroviruses. We paid attention to the effectiveness of the tested preparations and the type of antiviral mechanisms of antimicrobial action of plant antimicrobials against noroviruses.

Reviewer 2 Report

The manuscript reviewed anti-noroviruses activities of plant-original constituents and their potential usage in food industry. The manuscript was well written. This reviewer suggested accept after minor revision.

(1) The extract method referred in the manuscript are general method for extracting secondary metabolites from plant. This reviewer thinks providing some  specific method targeting antiviral constituents would be preferred.

(2) Table 1 listed components and their antimicrobial activity. Only antiviral activity would be preferred. 

Author Response

Dear Reviewer,

We would like to thank you very much for your insightful reviews of our manuscript. Thank you for all your comments and suggestions. Thank you for your taking the time.

The manuscript reviewed anti-noroviruses activities of plant-original constituents and their potential usage in food industry. The manuscript was well written. This reviewer suggested accept after minor revision.

(1) The extract method referred in the manuscript are general method for extracting secondary metabolites from plant. This reviewer thinks providing some specific method targeting antiviral constituents would be preferred.

Author response:

Section 2 (Methods of obtaining biologically active compounds from plant materials) has been removed as suggested by Reviewer 1.

(2) Table 1 listed components and their antimicrobial activity. Only antiviral activity would be preferred.

Author response:

We agree with this suggestion. Information on the antibacterial and antifungal activity of essential oils has been removed from Table 1. Table 1 in the current version contains information about noroviruses.

Reviewer 3 Report

The information is presented in a concise and organized way. All relevant bibliography has been consulted. Tables and the illustration are adequate. It is a  good review with relevant information about plant antiviral compounds and specifically against noroviruses. 

Author Response

Dear Reviewer,

We would like to thank you very much for your insightful reviews of our manuscript. Thank you for your taking the time.

Round 2

Reviewer 1 Report

This version of the manuscript is much better than previous one which was poorly prepared. Now authors focused just on noroviruses and the whole manuscript is much better. However, still there is more space for improvements.

First of all, native English speaker who is familiar with the topic of the manuscript had to check the whole manuscript because there are too much wrong constructions unnatural for English. The first one is obvious in a title. The structure ¨ “Potential possibilities “is unnatural for English and could not be used. My suggestion is to change the title into: „Antiviral potential of plants against noroviruses“ to better reflect the whole manuscript.

And please remove construction possible possibilities or potential possibilities from the rest of the manuscript.

Also, plant substances is kind of wierd construction. In a manuscript authors wrote about plant based extracts or plant metabolites. Please, use the word metabolites insted of substances... or just phytochemicals.

Also natural plant metabolites are chemicals so if authors reffer to syntetic chemicals they should use this construction not just chemicals. If they refer to plant-based chemicals they could use word phytochemicals or plant metbaolites. This should be uniformed in a manuscript.

I suggest the changes of the abstract into:

„Human noroviruses belonging to enteroviruses are one of the most common etiological agents of food-borne diseases. In recent years, intensive research has been carried out on the antiviral activity of plant substances that could be used for the preservation of fresh food because they are safer compared to syntetic chemicals. Plants preparations with proven antimicrobial activity, differ in their chemical composition, which significantly affects their biological activity.  Our review aimed to present the results of research related to the characteristics, applicability, and mechanisms of action of various plant based preparations and metabolites against norovirus. New strategies to combat intestinal viruses are necessary not only to ensure food safety and reduce infections in humans but also to lower the direct health costs associated with them.“

L 20-21. Please remove „Our review covers data from the scientific literature over the past decade“ or explain in a manuscript exactly how did you do literature serach, how many papers did you find and review. I think it is easier to remove this sentence.

The whole manuscript is review so please change title of the paragraph 5 into 5. Plant preparations as an antiviral agents agains noroviruses.

Please remove „This is a review of a large body of scientific research results from the last ten years.“ Or add dietals as I mention above.

Author Response

Dear Reviewer,

Thank you once again for your help, for all your comments and suggestions.

1. First of all, native English speaker who is familiar with the topic of the manuscript had to check the whole manuscript because there are too much wrong constructions unnatural for English. The first one is obvious in a title. The structure ¨ “Potential possibilities “is unnatural for English and could not be used. My suggestion is to change the title into: „Antiviral potential of plants against noroviruses“ to better reflect the whole manuscript.

Author response:

According to your suggestion, the manuscript text has been checked by the native English speaker Robin Royle.

The title of the article was also changed into: „Antiviral potential of plants against noroviruses“. Thank you for this suggestion.

2. And please remove construction possible possibilities or potential possibilities from the rest of the manuscript.

Author response:

It has been done.

3. Also, plant substances is kind of wierd construction. In a manuscript authors wrote about plant based extracts or plant metabolites. Please, use the word metabolites instead of substances... or just phytochemicals.

Also natural plant metabolites are chemicals so if authors reffer to syntetic chemicals they should use this construction not just chemicals. If they refer to plant-based chemicals they could use word phytochemicals or plant metabolites. This should be uniformed in a manuscript.

Author response:

This has been done. We used the word “metabolites” or “phytochemicals” instead of “substances”. Thank you for your explanation.

4. I suggest the changes of the abstract into:

„Human noroviruses belonging to enteroviruses are one of the most common etiological agents of food-borne diseases. In recent years, intensive research has been carried out on the antiviral activity of plant substances that could be used for the preservation of fresh food because they are safer compared to synthetic chemicals. Plants preparations with proven antimicrobial activity, differ in their chemical composition, which significantly affects their biological activity.  Our review aimed to present the results of research related to the characteristics, applicability, and mechanisms of action of various plant based preparations and metabolites against norovirus. New strategies to combat intestinal viruses are necessary not only to ensure food safety and reduce infections in humans but also to lower the direct health costs associated with them.“

Author response:

It has been done. Thank you for improving the abstract.

5. L 20-21. Please remove „Our review covers data from the scientific literature over the past decade“ or explain in a manuscript exactly how did you do literature search, how many papers did you find and review. I think it is easier to remove this sentence.

Author response:

Thanks for your suggestion. We agree with you. The sentence has been removed.

6. The whole manuscript is review so please change title of the paragraph 5 into 5. Plant preparations as an antiviral agents against noroviruses.

Author response:

The title of the paragraph 5 has been changed.

7. Please remove „This is a review of a large body of scientific research results from the last ten years.“ or add details as I mention above.

Author response:

The sentence has been removed.